# Unveiling the Potential of Natural Compounds: A Comprehensive Review on Adipose Thermogenesis Modulation

**DOI:** 10.3390/ijms25094915

**Published:** 2024-04-30

**Authors:** Jaeeun Shin, Yeonho Lee, Seong Hun Ju, Young Jae Jung, Daehyeon Sim, Sung-Joon Lee

**Affiliations:** 1Division of Biotechnology, College of Life Sciences and Biotechnology, Korea University, Seoul 02855, Republic of Korea; zzen2_97@naver.com (J.S.); dusgh48@naver.com (Y.L.); jooooo12345@naver.com (S.H.J.); lyjung51@gmail.com (Y.J.J.); tlaeogus369@naver.com (D.S.); 2Department of Food Bioscience and Technology, College of Life Sciences and Biotechnology, Korea University, Seoul 02855, Republic of Korea; 3Interdisciplinary Program in Precision Public Health, BK21 Four Institute of Precision Public Health, Korea University, Seoul 02846, Republic of Korea

**Keywords:** natural compound, adipocyte, browning, thermogenesis, BAT, brown adipose tissue, beige adipose tissue

## Abstract

The process of adipocyte browning has recently emerged as a novel therapeutic target for combating obesity and obesity-related diseases. Non-shivering thermogenesis is the process of biological heat production in mammals and is primarily mediated via brown adipose tissue (BAT). The recruitment and activation of BAT can be induced through chemical drugs and nutrients, with subsequent beneficial health effects through the utilization of carbohydrates and fats to generate heat to maintain body temperature. However, since potent drugs may show adverse side effects, nutritional or natural substances could be safe and effective as potential adipocyte browning agents. This review aims to provide an extensive overview of the natural food compounds that have been shown to activate brown adipocytes in humans, animals, and in cultured cells. In addition, some key genetic and molecular targets and the mechanisms of action of these natural compounds reported to have therapeutic potential to combat obesity are discussed.

## 1. Introduction

The prevalence of obesity has increased significantly over the past few decades, and it now ranks among the leading global causes of mortality and morbidity [1,2]. To combat obesity, which is frequently a risk factor for serious clinical conditions including cancer, heart disease, and diabetes [2], lifestyle modifications, such as reducing food intake or increasing activity, must be implemented. However, it is difficult to lose and maintain weight consistently through the implementation of these modifications alone; as a result, the development of medications or substances that can aid in either preventing or treating obesity is required.

Obesity is caused by increased adiposity, resulting in an accumulation of excess fat that could lead to the development of chronic diseases and a reduction in lifespan [2,3]. An essential function of adipose tissue is controlling energy balance. White adipose tissue (WAT; subcutaneous WAT, sWAT; inguinal WAT, iWAT), an anabolic fat, is essential for storing fat fuel in the form of triglycerides [4]. BAT, a catabolic fat with a high content of iron-containing mitochondria, is identified as producing heat when stimulated with cold or during exercise training [5,6]. These two stimuli are often highlighted due to their impact on BAT activity and metabolism. Exposure to cold has been the subject of the most extensive investigation as a method of initiating BAT activation, considering that BAT’s primary function in mammals is the regulation of non-shivering thermogenesis [7]. The exposure to cold temperatures triggers the activation of the sympathetic nervous system, prompting the release of norepinephrine, which activates the β-adrenergic receptor (β-AR) [8]. This activation initiates the cyclic adenosine monophosphate (cAMP)-dependent signaling pathways, leading to the increased uptake and oxidation of fuel for the generation of heat via uncoupling protein-1 (UCP1) [9]. UCP1 is a protein in the inner mitochondrial membrane and generates heat via a proton gradient [9]. BATs were originally considered to only be present in small rodents, contributing to the regulation of energy expenditure [7]. BATs are principally created in the fetus, and are known to vanish 10 years after birth, with only a small quantity of these BATs subsequently discovered in adults [10]. However, recent studies have revealed that adult humans have considerable quantities of BAT, identified through fluorodeoxyglucose-positron emission tomography–computed tomography (PET/CT) [11,12].

Furthermore, alongside conventional BAT, adults possess inducible BAT, termed beige adipocytes, which exhibit distinctive features that set them apart from WAT and BAT [13,14]. Adipocyte beiging, also known as the browning of WAT, is a physiological process wherein white adipocytes, traditionally recognized for their role in energy storage, adopt characteristics resembling BAT [15,16,17]. Adipocytes originate from mesoderm-derived mesenchymal stem cells (MSCs) and follow direct lineages, with white and beige adipocytes originating from different precursor cells to brown adipocytes [18,19,20]. The transcriptional regulation of adipogenesis involves two main phases: commitment and terminal differentiation [21]. While white, beige, and brown adipocytes have unique characteristics, they share common adipogenic machinery, such as CCAAT/enhancer binding protein (C/EBP) family members and peroxisome proliferator-activated receptor gamma (PPARG) [22,23]. During the commitment stage, C/EBP-β (CEBPB) and C/EBP-γ (CEBPG) are induced [24], leading MSCs to commit to the pre-adipocyte stage [25]. In the terminal differentiation phase, these factors activate PPARG and C/EBP-α (CEBPA) expression, facilitating the transition of pre-adipocytes to mature adipocytes. PPARG and CEBPA are pivotal in the regulation of adipogenesis, because their overexpression can prompt the differentiation of fibroblastic cell lines into adipocytes [26]. Brown and beige adipogenesis additionally involve specific transcription factors such as the PR-domain-containing 16 (PRDM16) and PPARG coactivator 1 alpha (PPARGC1A) [20]. The induction of beige adipocytes from both progenitor cells and mature whitened beige adipocytes represents a dynamic and adaptable mechanism for modulating adipose tissue function and energy homeostasis in response to various physiological and environmental cues [27].

Although the phenomenon of white fat browning in response to cold adaptation was first documented over three decades ago [28,29], recent investigations have delved deeper into the intricate molecular mechanisms and developmental lineages governing this process [14,30]. The natural compounds derived from foods or plants have become the backbone of the novel synthetic drugs currently used. For the last three decades, almost 50% of drug approvals have been compounds that were either indirectly or directly derived from natural compounds [31]. Therefore, natural food or plant compounds may also be promising therapeutics for obesity management through the promotion of adipocyte browning. In this review, our focus is on summarizing the information we have on 11 natural compounds (curcumin, berberine, resveratrol, ginsenoside, corylin, formononetin, phytol, luteolin, menthol, nobiletin, and rutin).

The selection process for these compounds began with a search strategy that employed a combination of keywords relating to natural compounds and adipose browning. In addition, an extensive review was completed of the existing literature concerning natural compounds and their effects on adipose tissue browning. Among numerous natural compounds, we reviewed the full texts of the selected articles to determine their eligibility for inclusion in the review. In particular, we selected those compounds which were the subject of extensive study in the past 5 years, or those compounds that provided promising results together with the possibility of further research. The compounds that showed promising results related to the induction of the browning of adipocytes in vivo and in vitro. Additionally, the results were required to exhibit beneficial metabolic effects for the compounds, thus rendering them potential candidates for obesity management. The diverse mechanisms of action and established safety profiles of the identified compounds made them particularly intriguing for further exploration in the context of promoting adipose tissue browning as a therapeutic strategy. This review will provide information on the beneficial metabolic effects of the identified compounds and highlight the potential suggested mechanisms. This review will provide information on the beneficial metabolic effects and highlight the potential suggested mechanisms of the selected compounds.

## 2. Natural Compounds

### 2.1. Curcumin

Curcumin (diferuloylmethane) is a turmeric-derived flavonoid [32]. It is often used as a spice in many Asian dishes, as well as a nutritional agent, a traditional Asian medicine, and a food preservative [32,33,34]. Along with these usages, knowledge about curcumin’s effect on the browning of adipocytes has been emerging, and the substance has been investigated in vivo and in vitro.

An in vitro experiment conducted by Lone et al. revealed that curcumin regulates the browning of adipocytes and stimulates lipid catabolism and mitochondrial biogenesis through the AMP-activated protein kinase AMPK-PGC1α-UCP1 pathway. In 3T3-L1 and primary inguinal white adipocytes from 10-day-old rats, the treatment with curcumin (20 µM) increased the mRNA and protein expression of *Ppargc1a*, *Pparg*, and *Ucp1*, the brown-adipocyte-specific genes. When co-treated with an adipocyte-browning cocktail consisting of rosiglitazone and triiodothyronine, the curcumin group also showed a significant increase in brown fat-specific genes including *Ppargc1a*, *Pparg*, PR-domain-containing 16 (*Prdm16*), and CCAAT/enhancer binding protein-ß (*Cebpb*), as well as transmembrane protein 26 (*Tmem26*), cell-death-inducing DFFA-like effector a (*Cidea*), fibroblast growth factor 21 (*Fgf21*), and Cbp/p300-interacting transactivator with Glu/Asp-rich carboxy-terminal domain 1 (*Cited1*) [35]. The up-regulation in these genes indicates a possible conversion of white to beige cells with curcumin treatment. In addition, curcumin treatment increased the protein expression levels of diacylglycerol choline phosphotransferase (Cpt1), cytochrome C, hormone-sensitive lipase (Hsl), and phosphorylated acetyl-CoA carboxylase (ACC). The up-regulated expression of Cpt1 and cytochrome C (CytC) indicated enhanced fat oxidation following curcumin treatment, and the elevated levels of Hsl and phosphorylated ACC reflected lipolysis and repressed fatty acid synthesis. Furthermore, the density of mitochondria was substantially elevated, which correlates with an increase in the protein level of Ppargc1a, showing that curcumin induces mitochondrial biogenesis. AMPK is a metabolic regulator that governs the utilization of fatty acids and the maintenance of energy balance [36]. Curcumin treatment significantly increased the phosphorylation of AMPK, leading to the hypothesis that curcumin induces the browning of adipocytes through the AMPK-mediated pathway. This study identified curcumin as a compound that stimulates lipid catabolism and mitochondrial biogenesis, and provides a unifying upstream mechanism for the activation of the AMPK-PGC1α-UCP1 pathway via curcumin [35].

Another in vitro study of Zhao et al. indicated that curcumin elevated energy expenditure and thermogenic markers through the regulation of Pparg. The treatment with curcumin induced the browning of 3T3-L1 adipocytes. The oxygen-consumption rate (OCR) of curcumin-treated (10 µM) adipocytes remained higher than that of adipocytes in the control group. Basal mitochondrial respiration in the curcumin-treated group was elevated by 16.5%, and maximal mitochondrial respiratory capacity was enhanced by 20.5% compared to the control group [37]. Since maximal OCR is an indicator that represents the ability of mitochondria to reserve energy [38], these results indicate that curcumin (10 µM) significantly promoted the mitochondrial respiration function. Furthermore, curcumin increased the protein levels of Ucp1, Prdm16, Ppargc1a, and Pparg in 3T3-L1-differentiated adipocytes, suggesting that curcumin regulates adipogenesis [37].

The in vivo applications of curcumin have been shown to reduce obesity and enhance the thermogenic capability of mice. In a study conducted by Zhao et al., the curcumin (50 mg/kg/day) treatment of diet-induced-obesity (DIO) mice displayed a decrease in WAT adipocyte sizes compared to the control group. Additionally, their BAT morphology demonstrated a higher number of lipid droplets, indicating that curcumin can inhibit the hypertrophy of white-fat cells and promote the transition into beige adipocytes. Furthermore, compared to the control group, the curcumin-fed DIO group showed increased WAT mRNA and protein-expression levels of Ucp1, Prdm16, Ppargc1a, and Pparg in BAT. Taken together, curcumin may offer a novel approach for regulating and treating obesity and other metabolic diseases through targeting PPAR-γ and its downstream genes [37].

In addition, Song et al. conducted an experiment to determine if dietary curcumin regulates both WAT inflammation and BAT-mediated energy expenditure. The curcumin intervention not only decreased macrophage infiltration in WAT but also shifted the composition of macrophage subtypes. Specifically, it reduced the proportion of M1-like macrophages, which are associated with pro-inflammatory responses, while increasing the proportion of M2-like macrophages, which are linked to anti-inflammatory actions and tissue repair. This suggests that curcumin not only mitigates WAT inflammation but also promotes a favorable shift in the balance of macrophage phenotypes towards a more anti-inflammatory profile, potentially contributing to improved metabolic health and adipose tissue function. In response to cold exposure (4 °C, 4 h), the curcumin intervention increased thermogenic capacity. The indirect calorimetric analysis showed that the curcumin intervention increased energy expenditure and core rectal temperature in response to cold exposure, and the thermogenic gene markers (*Ucp1*, *Ppargc1a*) in BAT were also elevated. Further findings revealed that curcumin promotes Ucp1 expression via PPARA and Pparg; however, PPARA and Pparg antagonists did not completely inhibit UCP1 expression. This indicates that there might be additional Pparg-independent pathways of curcumin in decreasing obesity and increasing thermogenesis [39].

Under cold exposure, curcumin has been shown to reduce body weight through enhancing thermogenesis in rodents [40]. The induction of the browning of adipose tissue using curcumin was reported by Wang et al., where curcumin (50, 100 mg/kg body weight) was orally administered to male C57BL/6 mice for 60 days. After 50 days of curcumin consumption, mice were exposed to a cold temperature (4 °C) for six hours. The curcumin-treated mice showed a decreased body weight and fat mass compared to untreated mice. In addition, when exposed to cold, the body temperature increased by 1 °C in the curcumin-treated group, indicating that curcumin increased thermogenesis upon cold exposure. The curcumin group also showed multilocular droplets depicting the browning of iWAT as well as an increase in fat-specific mRNA gene expressions (*Ucp1*, *Pgc1a*, *Prdm16*, iodothyronine deiodinase 2 (*Dio2*), *PPARA*, *Cidea*, ELOVL fatty acid elongase 3 (*Elovl3*), nuclear respiratory factor 1 (*Nrf3*), *and* ATP synthase (*ATPsyn*)) after cold exposure [40]. Norepinephrine, which interacts with ß3-AR in WAT, plays a crucial role in inducing the browning of WAT [36]. Curcumin supplementation increased plasma norepinephrine levels and induced ß3-AR expression in iWAT [40]. These observations suggest that curcumin can induce WAT browning through the norepinephrine-ß3AR pathway.

It has been demonstrated that curcumin not only showed efficacy in suppressing obesity in obese mice, but also indicated potential in regulating obesity in postnatal rats. Early postnatal overfeeding is known to cause metabolic imprinting, associated with decreased energy expenditure due to the elevated amount of WAT [41]. The relationship between curcumin and early postnatal overfeeding was reported by Zhu et al. In their study, starting from the third week of the weaning period, rat litters were supplemented with 2% curcumin for 10 weeks. Overall, the curcumin-treated rat litter showed a lower body weight and a higher lean mass ratio compared to the control group. In the group with 2% curcumin supplementation, increased thermogenesis along with browning-related *Ucp1* gene expression could be observed. Curcumin also increased ß3-adrenergic receptor mRNA expression in WAT, which was associated with an increase in norepinephrine levels in the serum. In summary, dietary curcumin in postnatal rats expedited energy expenditure, leading to decreased body weight, and mitigating metabolic diseases [42].

Imbalanced gut microbiotas are associated with obesity, and therefore alleviating the dysbiosis in gut microbiota is a frequent target in the treatment of obesity [43]. Previous research has shown that curcumin exerts beneficial effects by altering the gut-microbiome symbiosis and microbial bile acids (BAs) [44]. BAs are often recognized as signaling molecules that regulate metabolic homeostasis by activating Takeda G-protein-coupled receptor 5 (TGR5). The activation of TGR5 triggers mitochondrial uncoupling and enhances oxygen consumption through the up-regulation of the production of cAMP, activating the cAMP/protein kinase A (PKA) signaling pathway. This process results in increased mitochondrial activity, where mitochondria generate energy without producing ATP, leading to higher oxygen consumption. The up-regulation of cAMP and the subsequent activation of PKA play pivotal roles in mediating these effects, highlighting the significance of TGR5 signaling in regulating mitochondrial function and energy metabolism [45,46,47]. Han et al. examined the effects of the oral administration of curcumin on the gut microbiota, BA metabolism, and thermogenic adipose tissue. C57BL/6J mice, *Ucp1*^−/−^ mice, and *TGR5* knockout (*Gpbar1*^−/−^) mice were fed curcumin via oral gavage (100 mg/kg bodyweight) under a high-fat diet (HFD). Han et al. conducted 16S ribosomal DNA-sequencing on the fecal microbiota of curcumin-fed male C57BL/6J mice to examine the regulatory effects of curcumin on the gut microbiota. Under curcumin supplementation, the abundance of *Verrucomicrobia* and *Defferribacteres* increased while the relative abundance of *Bacteroidetes* decreased. In addition, curcumin treatment increased the relative abundance of *Lactobacillus*, a bile salt hydrolase (BSH)-activity-positive bacteria involved in BA metabolism, and *Clostridium*, which can convert primary BAs to the secondary BAs, deoxycholic acid and lithocholic acid [48]. These results indicate that the oral consumption of curcumin shaped the gut-microbiota community as well as ameliorating HFD-induced gut-microbiota dysbiosis. In addition, curcumin was shown to repress the expression of the BA synthesis key enzyme cytochrome P450 family 7 subfamily B member 1 (*Cyp7b1*), acting through an alternative pathway for BA synthesis for the improvement of metabolic dysfunction with the sacrifice of the classical pathway for BA synthesis. Furthermore, curcumin-reconstructed fecal microbiota transplantation also led to increased energy expenditure and Ucp1-dependent thermogenesis, which was depleted by abolishing the endogenous gut microbiota. These results show that curcumin supplementation can reshape the gut microbiota to alleviate obesity and increase thermogenesis [49].

Despite its beneficial effects, some research has shown that higher doses of curcumin can lead to adverse effects. Higher doses (20 and 30 µM) of curcumin on 3T3-L1-differentiated adipocytes inhibited the mitochondrial respiratory function [37]. Previous research has demonstrated that some natural compounds exhibit a non-monotonic dose curve, which may be one of the reasons for such adverse effects [41]. Therefore, further research is needed to determine the appropriate concentration of curcumin for beneficial applications in humans.

In all, both in vitro and in vivo studies, curcumin has demonstrated promising effects in regulating adipocyte browning, enhancing lipid catabolism, and promoting mitochondrial biogenesis through various pathways. It activates the AMPK-PGC1α-UCP1 pathway and influences Pparg, leading to increased energy expenditure, thermogenic markers, and reduced obesity in mice. Curcumin also shows potential in regulating gut microbiota to alleviate obesity. Nevertheless, the mechanisms connecting these outcomes to the gut microbiota and determining the source of the increased norepinephrine induced with curcumin are yet to be elucidated (Figure 1). Moreover, identifying the precise metabolic action mechanism of curcumin poses challenges due to its notably low systemic bioavailability. Furthermore, caution is advised regarding the dosage of curcumin, as higher concentrations may have adverse effects on mitochondrial respiratory function. More research is needed to establish the optimal curcumin concentration for beneficial applications in humans, considering the dose–response curve observed in some natural compounds.

### 2.2. Berberine

Berberine is a natural compound isolated from the plant *Rhizoma Coptidis* [50] and is known to have anti-diabetic, anti-cancer, and anti-hyperlipidemic effects in humans [50,51,52,53]. The previous findings have suggested that berberine may also elevate thermogenesis in mature adipocytes due to its anti-obesity effects. It has been shown that berberine accelerates the browning process of adipocytes and increases thermogenesis and energy expenditure in cells.

Berberine activates the AMPK signaling pathway targeting several downstream factors such as sirtuin 1 (SIRT1), Ppargc1a, and Fgf21 to display its functions [54,55,56]. Berberine up-regulates thermogenic capacity in vitro through the activation of the AMPK-SIRT1 pathway. Studies by Xu et al. examined whether berberine induces the browning process in 3T3-L1 white preadipocytes and H1B1b brown preadipocytes [54]. Berberine (5 μM) exhibited the characteristics of brown-like adipocytes with increased mitochondrial density and smaller lipid droplets. In 3T3-L1 cells, berberine up-regulated the SIRT1 and UCP1 protein level along with the phosphorylation of AMPK in a dose-dependent manner [54]. Previous studies have demonstrated that Sirt1 plays an important role in energy metabolism and the remodeling of adipose tissues [57,58,59]. Since Sirt1 is a downstream target of AMPK, Xu et al. postulated that berberine activates the AMPK-SIRT1 signaling axis. Silencing SIRT1 abolished berberine’s effect, suggesting an AMPK-SIRT1-dependent mechanism in regulating thermogenesis. This research concluded that berberine elevated thermogenesis, accompanied by altercations in the AMPK-SIRT1 pathway; however, the specific downstream mechanisms were unclear [54].

Further insights into berberine’s mechanisms of action in vitro were provided in the work of Zhang et al. [55]. In C3H10T1/2-derived BAT, berberine increased the mitochondrial mass. Furthermore, berberine elevated the levels of *Ppargc1a* and *Ucp1* mRNA and the silencing of *Ppargc1a* negated *Ucp1* induction by berberine, showing that berberine affected *UCP1* activity in a *PGC1α*-dependent manner. Furthermore, the silencing of AMPK reduced berberine’s effect on UCP1 and PGC1α protein-level induction, demonstrating that the AMPK-PGC1α pathway is a major mechanism of berberine’s effects [55].

FGF21 is a member of the FGF family and plays a significant role in regulating lipid homeostasis and peripheral glucose-tolerance levels [60]. The increased FGF21 levels in BAT show enhanced energy expenditure [61]. In addition, berberine increased Fgf21 protein expression, a key player in lipid homeostasis, through an AMPK-dependent mechanism in C3H10T1/2 BAT. In addition, berberine increased thermogenic gene expressions (*Ucp1*, *Cidea*, and *Cpt1b*), displaying enhanced energy expenditure [56].

Human studies, including on overweight patients with non-alcoholic fatty liver disease (NAFLD) and subsequent investigations on DIO and lean mice, have highlighted berberine’s potential to increase the thermogenic activity of BAT. Wu et al. conducted a study with 10 overweight NAFLD patients. They consumed 0.5 g of berberine three times a day for a month before undergoing cold-activated PET/CT scanning. After one month of berberine intervention, the volume and activity of BAT and the basal oral temperature all increased from 36.3 °C to 36.5 °C (all *p* < 0.05), showing signs of increased thermogenic capacity [62].

In DIO mice, 6 weeks of berberine treatment increased the rectal temperature during cold exposure at 4 °C and increased BAT mass and brown adipocyte-related gene expressions (*Ucp1*, *Prdm16*, *Cidea*, *Ppargc1a*, cytochrome c oxidase subunit 7A1 (*Cox7a1*), cytochrome c oxidase subunit 8b (*Cox8b*), and *Dio2*) [62]. Berberine treatment reduced body weight and increased energy expenditure, thermogenesis, and BAT activity in obese db/db mice. Correspondingly, the gene expression of browning factors including *Ucp1*, *Ppargc1a*, and nuclear respiratory factor 1 (*Nrf1*) was augmented according to berberine treatment in a concentration-dependent manner. However, these effects were only found at lower temperatures, whereas thermoneutral conditions blocked thermogenesis by berberine [62].

The effects of berberine on thermogenesis were further explored in both BAT and WAT in vivo, revealing increased energy expenditure and adaptive thermogenesis through the comprehensive lab-animal-monitoring system (CLAMS) [55]. The balance between energy expenditure and caloric intake is a key factor in controlling energy homeostasis [63]. Through CLAMS, the berberine group showed a higher rate of oxygen consumption and CO_2_ production with an additional increase in the whole-body energy expenditure [55]. Since the increase in energy expenditure was not due to an increase in physical activity, the respiratory exchange ratio [64] indicated the proportion of energy derived from carbohydrates as opposed to lipids. The substantial reduction in respiratory exchange ratio (RER) followed by berberine treatment indicates that berberine alters the fuel preference in favor of fatty acids. Furthermore, through ^18^F-FDG with micro-PET/CT, the activity of BAT increased in the berberine group. In addition, with the decrease in size of lipid droplets in BAT and WAT, the expression of thermogenic genes including *Ucp1*, *Cidea*, *Cox8b*, and perilipin 5 (*Plin5*) were strongly up-regulated in BAT. These results support the contention that berberine could activate the thermogenic programming of BAT and WAT in mice [55].

In line with previous research, Xu et al. also investigated the mechanism of berberine on adipose tissue remodeling in DIO C57BL/6J mice. In their results, berberine supplementation (25 and 100 mg/kg bw) increased the oxygen consumption rate and CO_2_ production rate, suggesting increased energy metabolism. In addition to the decreased body weight in the berberine group, micro-CT scans and 3D reconstructions of the intrascapular regions of BAT and WAT revealed that berberine increased the content of BAT while decreasing the content of WAT. Under cold exposure at 4 °C for 2 h, berberine supported the maintenance of the body core temperature through maintaining the rectal temperature higher by 3 °C compared to the control group. In addition, the relative gene and protein expression of brown fat-enriched genes in WAT and BAT were shown to increase (*Ppargc1a*, *Ucp1*, *Cidea*, *Prdm16*, *Dio2*, and *Cox7a1*), along with fatty acid-oxidation genes (fatty acid-transport protein 1 (*Fatp1*) and carnitine palmitoyltransferase 1 (*Cpt1*)). These results demonstrate that berberine promotes WAT browning and the thermogenic activity of BAT in DIO mice [54].

In summary, the different roles of berberine as a comprehensive modulator of thermogenesis, adipose tissue remodeling, lipid homeostasis, and in energy expenditure have been highlighted. However, further research, including clinical trials, is needed to validate and fully understand the mechanisms involved and to determine the optimal conditions for its use in humans. Berberine has been demonstrated to alleviate adipose browning via the AMPK pathway (Figure 2). However, previous research has suggested that the hypothalamic ventromedial hypothalamus (VMH) AMPK might play a role in BAT thermogenesis and the browning of WAT [65,66]. Similarly, the involvement of other hypothalamic nuclei in the thermogenic response induced by berberine cannot be discounted and warrants further investigation. Additionally, the effects of berberine on the alternative to additional progenitor cell types being committed to brown fat-like cells require clarification through fate-mapping experiments. In essence, berberine could be a promising therapeutic candidate for addressing metabolic disorders and obesity. The intricate interplay with the key signaling pathways of berberine (AMPK, SIRT1, and FGF21) opens avenues for the further exploration of its therapeutic potential in addressing metabolic disorders and obesity.

### 2.3. Resveratrol

Resveratrol (3,5,4’-trihydroxystilbene) is a natural polyphenol [67] produced by a variety of plants (grapes, raspberries, blueberries, etc.) when attacked by pathogens [68,69]. In both human and animal models, resveratrol has been proven to provide health benefits, including a lower risk of obesity-related diseases [70,71,72,73].

In their investigation, Wang et al. examined the effects of resveratrol on differentiated inguinal WAT primary cells. The application of resveratrol (20 and 40 µM) significantly inhibited lipid accumulation and decreased the mRNA expression of adipogenic markers *Pparg* and the *fatty acid-binding protein* (*aP2*) in a dose-dependent manner. Even though a low concentration of resveratrol (10 µM) showed no effect on decreasing lipid accumulation, it elevated the expression of brown-adipocyte-specific genes (*Prdm16*, *Ucp1*, *Cidea*, *Elovl3*, *Ppargc1a*, TNF receptor superfamily member 9 (*Tnfrsf9*), T-vox transcription factor 1 (*Tbx1*), and transmembrane protein 26 (*Tmem26*)). Since the browning of WAT is strongly associated with an elevated cellular respiration level, resveratrol also showed an increase in basal oxygen consumption by 1.6-fold compared to the control. In addition, resveratrol also increased the phosphorylation of AMPK and the expression of Sirt1; however, the knockdown of AMPK resulted in the ablation of Sirt1 expression and the phosphorylation of AMPK, along with the down-regulation of the expression level of thermogenic genes [74]. This shows that resveratrol induces the browning of WAT in an AMPK-SIRT1-dependent manner.

The mammalian target of rapamycin (mTOR) is widely recognized as a key regulator of various cellular metabolic processes [75]. Previous research has shown that resveratrol influences mTOR complex 1 (mTORC1) or its downstream site P70S6 [76]. In 3T3-L1 adipocytes, resveratrol treatment increased the gene expression of key thermogenic BAT-linked markers (*Ucp1*, *Ppargc1a*, and *Pparg*). Furthermore, an increased level of Ucp1 expression upon resveratrol treatment was also demonstrated through immunostaining at the cellular level. To elucidate the mechanism of resveratrol on adipocyte browning, Liu et al. focused on mTORC1, and showed that treatment with an mTORC1 inhibitor (rapamycin, 10 μM) eliminated all the effects of resveratrol. These results suggest that resveratrol induces the browning of adipocytes through the mTOR-UCP1 pathway [77].

To further analyze the browning effects of resveratrol on iWAT in vivo, CD1 female mice were fed an HFD with or without 0.1% resveratrol. The resveratrol inhibited weight gain and elevated mRNA expression levels of thermogenic genes (*Ucp1*, *Prdm16*, *Ppargc1a*, pyruvate dehydrogenase (*PDH*), and *CytC*). Additionally, the increase in the oxygen-consumption rate due to resveratrol supplementation indicated that resveratrol promotes lipid oxidation. Furthermore, the resveratrol-treated mice showed higher Ucp1 and Prdm16 protein levels, which were accompanied with an elevated level of phosphorylated AMPK. These results demonstrate that resveratrol induces adipocyte browning in an AMPK-Ucp1-dependent manner [74].

Maternal obesity impairs the function of brown adipocytes and increases the chances of obesity in offspring [78,79]. Maternal female C57BL/6J mice were fed an HFD either with or without a supplement of 0.2% resveratrol. Similarly, male and female offspring were fed the same diet. The supplementation of resveratrol increased multilocular adipocytes and brown adipocyte-related genes (*Ucp1*, *Prdm16*, *Cidea*, *Ppargc1a*, and *Pparg*). Resveratrol supplementation was also shown as preventing the protein levels of Ucp1, Prdm16, Sirt1, and phosphorylated AMPKα levels from decreasing due to HFD. These findings collectively suggest that when administered to maternal mice, resveratrol enhances the thermogenic function of BAT in offspring exposed to HFD. Additionally, energy expenditure and insulin sensitivity increased under resveratrol supplementation, indicating the enhanced browning of adipocytes in the offspring. In conclusion, this study demonstrates that the resveratrol supplementation of mice during pregnancy promotes BAT recruitment as well as beige adipocyte development in the offspring [80].

Considering the limited absorption rate of polyphenolic compounds, an increasing number of research findings have suggested that resveratrol may primarily mitigate obesity and its associated health conditions through the modification of the composition of the gut microbiota [81,82,83]. BAs, products of cholesterol, are secreted to the duodenum from the gall bladder during digestion and facilitate the absorption of lipophilic nutrients such as lipids and vitamins [84]. Hui et al. aimed to clarify the potential role of resveratrol in modulating BAs in the gut to increase energy expenditure and promote WAT beiging. The administration of 0.4% resveratrol for 10 weeks on db/db mice improved glucose metabolism, which is linked to BAT activation and WAT browning. The resveratrol supplementation showed no effect on body weight; however, fasting glucose and insulin levels were reduced, and glucose intolerance was alleviated under resveratrol supplementation. While the weight of BAT increased, the weight of WAT decreased. Additionally, morphological changes in BAT and iWAT into more brown-adipose-like lipid droplets were observed. Furthermore, the mRNA expression of thermogenic genes (*Ucp1*, *Cidea*, *Prdm16*, and *Ppargc1a*) was also up-regulated in BAT and iWAT under resveratrol supplementation. In addition, the administration of resveratrol reorganized the gut microbiota, measured by an increase in the concentration of lithocholic acid in the plasma and feces [85].

Since lithocholic acid is a ligand for TGR5, the authors proposed that resveratrol’s ability to enhance BAT activity and WAT browning was partly explained by the increase in lithocholic acid. To elucidate whether the gut microbiota is an essential factor for resveratrol’s beneficial effects, the researchers continued with an antibiotic treatment on db/db mice. With the subsequent absence of gut microbiota, all effects of resveratrol were ablated, identifying the connection between resveratrol and gut microbiota. In addition, fecal microbial transplantation from the resveratrol-treated mice to the control mice revealed an increase in the mRNA expression level of *Ucp1* and an increase in glucose homeostasis. In summary, resveratrol supplementation promotes adipocyte browning and BAT activation by way of the gut microbiota–bile acid–TGR5-UCP1 pathway [85].

In conclusion, resveratrol, a natural polyphenol found in various plants, has demonstrated its potential in combating obesity and related health issues. Several studies have shown that resveratrol promotes the browning of white adipocytes and enhances energy expenditure. It achieves these effects through multiple pathways, including the activation of SIRT1, AMPK, and mTOR, which are involved in regulating thermogenesis and lipid metabolism (Figure 3). Resveratrol supplementation has been linked to increased BAT (BAT) activity and the development of beige adipocytes, which contribute to an improved energy balance. Furthermore, the impact of resveratrol extends beyond individual health, as it has been associated with maternal programming effects that benefit offspring through the promotion of brown fat development. Additionally, resveratrol’s influence on the gut microbiota and bile acid metabolism have further contributed to its anti-obesity effects. Overall, resveratrol may offer a promising natural approach to combat obesity and its associated metabolic disorders, and it achieves these effects through a variety of interconnected mechanisms. However, despite extensive research on resveratrol both in vitro and in vivo, this mechanism of action remains poorly understood across various conditions and doses. While numerous effects have been elucidated in vitro and in vivo, many of these have not been subsequently translated to clinical studies. This discrepancy is primarily attributed to resveratrol’s pharmacokinetics. Although orally absorbed in humans (approximately 70%), its systemic bioavailability is markedly low (approximately 0.05%) [86,87,88]. While more studies, such as clinical trials, are needed to fully understand its long-term effects and optimal dosages, the findings underscore the potential of resveratrol as a novel anti-obesity or thermogenic compound.

### 2.4. Ginsenosides

Ginsenosides are natural steroid-like saponins particularly found in *Panax ginseng*, *Panax pseudoginseng*, and *Codonopsis pilosula* [89,90]. There are over 30 different types of ginsenosides, broadly categorized into two main groups, 20(S)-protopanaxadiol (PPD) and 20(S)-protopnanxatriol (PPT) [91]. The PPDs carry a carboxyl group at the C-6 position, whereas the PPTs do not [89]. Ginsenosides are commonly used in Chinese and Korean medicine due to their anti-oxidative, anti-cancer, and anti-aging properties [92,93,94]. Recently, research has also highlighted the potential of ginsenosides in promoting weight loss and increasing insulin sensitivity.

Due to ginsenoside Rb1′s anti-obesity effects, Mu et al. specifically focused on examining adipocyte browning and thermogenesis. Differentiated 3T3-L1 adipocytes were treated with various concentrations of ginsenoside Rb1 (0.1–100 μM), resulting in a dose-dependent increase in basal glucose uptake under ginsenoside Rb1 treatment. Moreover, ginsenoside Rb1 (10 μM) increased the mRNA expression levels of thermogenic genes (*Ucp1*, *Ppargc1a*, and *Prdm16*) in differentiated 3T3-L1 adipocytes compared to the control. Ginsenoside Rb1 also demonstrated an increase in Pparg activity, which was counteracted with GW9662, a PPARG antagonist. These results suggest that ginsenoside Rb1 stimulates the expression of specific genes related to fat browning through the induction of Pparg, as observed in vitro [95].

Despite ginsenoside Rb1′s ability to increase both the browning of WAT and energy expenditure, Lim et al. conducted further studies to investigate the specific mechanism underlying these abilities of ginsenoside Rb1. Rb1 was found to decrease the size and morphology of lipid droplets in 3T3-L1 cells and increase the phosphorylation of the liver’s kinase B1 (Lkb1)-AMPKα1/2-acetyl-CoA carboxylase (ACC) pathway. Notably, the examination of the impact of Rb1 on the AMPKα1/2 pathway revealed an up-regulation of sirtuins (Sirt1, Sirt3), closely associated with AMPKα1/2 function. Additionally, treatment with ginsenoside Rb1 led to an increase in the expression of BAT-related genes (*Ucp1*, *Ppara*, and *Ppargc1a*) as well as beige-specific genes (*Prdm16*, *Tmem26*, *Tbx1*, and *Tnfrsf9*), indicating its potential for inducing thermogenic capacity [96].

Ginsenoside Rb2 has been reported to decrease hepatic lipid accumulation in DIO mice and to reduce triglyceride levels in differentiated 3T3-L1 mature adipocytes [97,98]. While ginsenoside Rb2 has shown promise as a potential anti-obesity compound, its effects on brown and beige adipose tissues were insufficiently elucidated. Hong et al. investigated the effects of Rb2 on differentiated 3T3-L1, BAT-differentiated C3H10T1/2, and iWAT primary cells. The treatment with Rb2 (10 μM) induced the activation of brown fat and the browning of white fat via the stimulus of the expression of uncoupling protein-1 (Ucp1) at both mRNA and protein levels. Moreover, Rb2 increased the expression of thermogenic and mitochondrial genes (*Prdm16*, *Ppargc1a*, *Cidea*, *Dio2*, *Ucp1*, *Fgf21*, *ATP syn*, and *Cox8b*). Additionally, Rb2 treatment led to an increase in the phosphorylation level of AMPK, along with its downstream thermogenic genes, *Ppargc1a* and *Ucp1*. Notably, the inhibition of AMPK using compound C diminished the induction of AMPK phosphorylation and downstream thermogenic genes, suggesting that Rb2 acts through the AMPK-Ppargc1a-Ucp1 pathway to induce the browning of WAT [99].

To elucidate the function of Rb2 in vivo, the metabolic performance was assessed on DIO mice supplemented with Rb2. Following 9 weeks of supplementation with Rb2 (40 mg/kg/day), significant improvements were observed in the DIO mice. Rb2 efficiently decreased body weight, enhanced insulin sensitivity, and stimulated energy expenditure. The histological and gene-expression analyses revealed that Rb2 induced the activation of brown fat and the browning of white fat by reducing lipid droplets. Furthermore, Rb2 increased the expression of Ucp1 at both mRNA and protein levels. Notably, Rb2 promoted AMPK phosphorylation both in vitro and in vivo, demonstrating AMPK-dependent effects. The specific inhibition of AMPK prevented the promotion of Ppargc1a and Ucp1 expression induced by Rb2, further supporting its AMPK-dependent mechanism of action. Overall, Rb2-activated brown fat and the promotion of browning in white fat led to increased energy expenditure and thermogenesis, thereby alleviating obesity and metabolic diseases. These findings suggest that Rb2 holds promise as an effective treatment for obesity [99].

Ginsenoside Rg3 is known to possess diverse therapeutic effects, including anti-obesity properties. Kim et al. demonstrated a novel role for Rg3 in inducing the browning of mature 3T3-L1 adipocytes through the up-regulation of the expression of browning-related genes (*Ucp1*, *Prdm16*, *Ppargc1a*, *Cidea*, and *Dio2*). Additionally, Rg3 was found to induce the expression of genes associated with beige adipose tissue (*Tnfrsf9* and *Tmem26*) and lipid metabolism (fatty acid synthase (*Fasn*), sterol regulatory element-binding protein 1 (*Srebp1*), and medium-chain acyl-CoA dehydrogenase (*Mcad*)), thereby activating lipid metabolism. Moreover, Rg3 inhibited the accumulation of lipid droplets and reduced their size in mature 3T3-L1 adipocytes. Previous studies have suggested that Rg3 exhibits anti-obesity effects through the activation of AMPK [100]. Kim et al. further investigated the browning effect of Rg3 through AMPK signaling. They found that Rg3 dose-dependently increased the phosphorylation of AMPK and its downstream targets (Ucp1, Ppargc1a), which were abolished by treatment with compound C, an AMPK inhibitor. These results demonstrate the anti-obesity efficacy of Rg3 through the activation of AMPK, leading to a browning response in 3T3-L1 cells [100].

In summary, ginsenosides, including Rb1, Rb2, and Rg3, demonstrate promising anti-obesity effects by promoting the browning of WAT, enhancing thermogenesis, and stimulating adipocyte browning. Rb1 activates browning-related genes and shows potential for thermogenesis by inducing PPARG expression. Its mechanism involves the activation of the Lkb1-AMPK1/2-ACC pathway and an increase in sirtuins (Sirt1 and Sirt3), contributing to its thermogenic capacity. Rb2 activates BAT through the AMPK-Ppargc1a-Ucp1 pathway, promoting the browning of WAT and effectively reducing body weight in DIO mice. Furthermore, Rb2 improves insulin sensitivity and enhances energy expenditure. Rg3, via the AMPK signaling pathway, up-regulates the expression of browning-related genes, activates lipid metabolism, and reduces lipid-droplet accumulation in mature adipocytes (Figure 4). However, further investigation is needed to elucidate whether other ginsenosides (Rb1, Rg1, Rg5, and Re) also contribute to the browning process. Furthermore, while compound C is frequently utilized as an AMPK inhibitor to evaluate alterations in the AMPK signaling pathway, it possesses limitations. Therefore, additional analyses are warranted to comprehensively identify the mechanisms of ginsenoside in adipocytes. Overall, ginsenosides contribute to the promotion of adipose tissue browning, the stimulation of thermogenesis, and the improvement of positive metabolic pathways, offering potential therapeutic options for obesity and metabolic disorders.

### 2.5. Corylin

Corylin, a natural flavonoid compound derived from the fruit of *Psoralea corylifolia* L. [115], has garnered attention for its therapeutic potential in various metabolic disorders. Among its notable effects, corylin stimulates osteoblast proliferation, making it a promising candidate for osteoporosis treatment [116,117]. Additionally, its benefits extend to addressing type II diabetes, obesity-induced fatty liver disease, and atherosclerosis [118,119,120,121]. Despite these known effects, the precise mechanisms underlying corylin’s actions on adipose browning remain elusive.

In a study conducted by Chen et al., the focus was on elucidating corylin’s anti-obesity effect and its impact on adipocyte browning. The research utilized the 3T3-L1 cell line and a DIO mouse model. Over a nine-week period of corylin administration, significant reductions in visceral fat mass and the size of WAT adipocytes were observed. In addition, in the DIO–corylin group, an increase in energy expenditure showed corylin’s lipolytic ability to reduce adipocyte size, or, in other words, to promote adipocyte browning. The experiments in vitro were performed to determine the mechanism underlying corylin’s thermogenic activity. According to previous research, Sirt1 has the ability to stimulate thermogenesis and decrease body fat [122,123,124,125]. Chen et al. discovered that corylin has a high affinity for binding to SIRT1 and ß3-AR through hydrogen bonding. The researchers concluded that corylin exerted browning and lipolysis effects via SIRT1- and ß3-AR-dependent pathways via the activation of SIRT1 and ß3-AR in vitro [101].

In summary, corylin, which is currently attracting research attention as a natural flavonoid with various therapeutic effects, can also mediate browning and lipolysis effects through the SIRT1 and ß3-AR pathways (Figure 4). While these findings have shed some light on some potential mechanisms, it is also acknowledged that more extensive research is needed to unravel the complex pathways through which corylin combats obesity and promotes adipocyte browning. The identification of its binding affinities and inhibitory effects on key regulators opens avenues for further exploration, emphasizing the need for a comprehensive understanding of corylin’s therapeutic potential in metabolic disorders.

### 2.6. Formononetin

Formononetin (7-Hydroxy-4′-methoxyisoflavone) is a natural isoflavone found in red clover (*Trifolium pratense*) and *Astragalus membranaceus* [126]. It has been used as a traditional Chinese herbal medicine for hundreds of years to maintain daily health. Formononetin is known to be endowed with antioxidants and anti-inflammatory and anti-cancer activities [127,128,129,130].

In isolated primary inguinal adipocytes differentiated into mature adipocytes, formononetin (1 μM) was shown to increase the mRNA expression level of thermogenic genes (*Ucp1*, *Ppargc1a*, *Prdm16*, *Cidea*, and *Dio2*), along with an elevated basal and uncoupled mitochondrial respiration rate. In addition, formononetin demonstrated a synergistic increase in the expression of thermogenic genes with thermogenic inducers (AM580, RAR agonist; CL-316,243, β3-adenoreceptor agonist; bexarotene, RXR agonist; and WY-14643, PPARA agonist). As formononetin dose-dependently elevated the luciferase activity of Pparg, the silencing of Pparg negated the effects of formononetin in elevating the thermogenic capacity of differentiated adipocytes. Based on these results, formononetin was shown to manage adipocyte thermogenesis through the Pparg-Ucp1 pathway [102].

Formononetin (50 mg/kg/day) was demonstrated to decrease the body weight of diet-induced obese mice when they were provided with a daily supplement for 8 weeks. A decrease in fat mass was also found, with lean mass being unaltered. The adipocytes not only showed a more multilocular structure, but also the oxygen-consumption rate increased. Furthermore, the mRNA expression level of *Ucp1*, *Ppargc1a*, *Prdm16*, *Cidea*, and *Dio2* increased in the formononetin group compared to the control group. These results convey the effect of formononetin in decreasing obesity through the augmentation of adipocyte browning and BAT activation [102].

Overall, formononetin, a natural compound identified in soy-based foods, demonstrates anti-obesity effects. In isolated adipocytes, it boosts thermogenic gene expression and mitochondrial activity via the Pparg-Ucp1 pathway (Figure 4). In obese mice, formononetin supplementation reduced body weight and fat mass, and enhanced oxygen consumption, suggesting its potential in combating obesity via the promotion of adipocyte browning and BAT activation both in vivo and vitro. However, the studies of its anti-obesity effects have focused more on its results in attenuating body weight and visceral fat accumulation under HFD [131]. More studies must be performed to identify the thermogenic or browning capacity of formononetin and its underlying mechanisms.

### 2.7. Phytol

Phytol, a branched-chain fatty alcohol abundant in nature as a component of the chlorophyll molecule, is released during the ruminal digestion of green vegetation in ruminant animals [132,133]. It is present in a number of foods, particularly in milk fat and marine foods [134]. Phytol’s metabolite, phytic acid, has been identified as inducing browning in adipocytes; additionally, phytol and its metabolites have been shown to activate PPARA [135]. Dietary supplementation with phytol has been shown to mitigate obesity and enhance the expression of PPARA target genes, indicating its role in activating PPARA in adipose tissues [136].

In differentiated 3T3-L1 cells, phytol (100 μM) increased the mRNA expression of Ucp1 along with brown adipocyte-marker genes (*Prdm16*, *Ppargc1a*, *Cidea*, and *Elovl3*), and beige adipocyte-specific genes (*Tnfrsf9* and *Tmem26*). Additionally, phytol elevated the protein levels of *Ucp1*, *Prdm16*, *Ppargc1a*, *Pdh*, and *CytC*, and increased mitochondrial content and oxygen-consumption rate. Phytol also phosphorylated AMPK1/2, accompanied by an increase in the protein levels of brown adipocyte-specific genes (*Ucp1*, *Prdm16*, *Ppargc1a*, *Pparg*, and *CytC*). However, this effect of phytol was abolished with the addition of an AMPK inhibitor, compound C. These results suggest that phytol stimulates adipocyte browning in vitro through the AMPK1/2 signaling pathway [103].

In 3T3-L1 preadipocytes, phytanic acid (50 μM) promoted beige adipogenic differentiation and up-regulated the mRNA levels of brown adipogenic markers (*Ppargc1a*, *Prdm16*, *Ucp1*, *Ppara*, and *Cidea*). However, in uncommitted C3H10T1/2 cells, phytic acid did not alter brown-like adipogenic differentiation or mRNA-expression levels of adipogenic regulators (*Prdm16* and *Ucp1*). Phytic acid induced beige/brown adipogenesis in a PPARA-dependent manner, along with increased mitochondrial biogenesis and oxygen consumption, which was inhibited by co-treatment with GW6471, a PPARA antagonist. These findings suggest that phytanic acid promotes beige adipogenic differentiation, with PPARA being a key regulator [104].

In 5-week-old C57BL/6J male mice, phytol (500 mg/kg) was shown to inhibit body-weight gain and decreased the amount of iWAT compared to controls on an HFD. Furthermore, iWAT from the phytol group exhibited multilocular droplets compared to controls, indicating that phytol stimulated the browning of iWAT. Additionally, the phytol group showed an elevated expression of brown adipocyte-marker genes (*Ucp1*, *Prdm16*, *Pgc1a*, *Pdh*, and *CytC*) along with the activation of the PRKAA1/2 signaling pathway [103]. These results suggest that phytol stimulates the browning of WAT in mice through PRKAA1/2 activation and increased brown adipocyte-marker genes.

In summary, phytic acid, a metabolite of phytol, has demonstrated its potential for inducing browning in adipocytes and enhancing thermogenesis. It activates key regulatory pathways such as PPARA and PRKAA1/2 in adipose tissues. The studies using various cell models have shown that phytol and its derivatives elevated the expression of brown and beige adipocyte-marker genes, including *Ucp1*, *Prdm16*, and *Ppargc1a*, thereby promoting the browning process. Moreover, in vivo experiments involving mice receiving orally administered phytol showed inhibited body-weight gain and decreased the iWAT index and the formation of multilocular droplets in iWAT, indicative of stimulated browning. These effects were associated with an elevated expression of brown adipocyte-marker genes and the activation of the PRKAA1/2 signaling pathway (Figure 4). Overall, these findings collectively highlight the potential of phytol and its metabolites as candidates for combating obesity and related metabolic disorders through the promotion of adipocyte browning and increased thermogenic activity.

### 2.8. Luteolin

Luteolin, a naturally occurring flavonoid found abundantly in various edible and medicinal plants such as carrots, celery, olive oil, oregano, and thyme [137], has demonstrated promising effects in mitigating diet-induced obesity and insulin resistance in C57BL/6 mice [138,139].

In 3T3-L1 adipocytes, luteolin significantly increased the expression of gene and protein levels involved in lipid metabolism and browning (*Ucp1*, *Ppargc1a*, *Sirt1*, and *Tfam*), along with increasing the phosphorylation levels of AMPK and ACC. Additionally, luteolin decreased the mRNA and protein levels of Srebp1-c, Cebpa, and Pparg, showing the ability to inhibit adipogenesis. Luteolin also reduced intracellular lipid accumulation, suggesting its potential role in reducing lipid storage in adipocytes. These findings suggest that luteolin has the potential to regulate lipid metabolism and induce browning in adipocytes [105]. However, the studies in this field are currently limited to in vitro cellular experiments, and there are still many unknowns regarding the action mechanisms and the most relevant receptors in vivo. Further extensive investigation is required to clarify these aspects.

Further investigation by Zhang et al. revealed that dietary supplementation with 0.01% luteolin in C57BL/6 mice fed either a low-fat diet or an HFD led to increased energy expenditure, as evidenced by elevated CO_2_ production, oxygen consumption, and respiratory exchange ratio. The mice supplemented with luteolin exhibited denser small BATs compared to the controls, along with increased mRNA expressions *of Ppargc1a*, *PPARA*, *Cidea*, and *Sirt1*. The increased expression of genes related to the AMPK-Ppargc1a signaling axis confirmed luteolin’s participation in the AMPK-Ppargc1a pathway through in vivo experiments. Similar effects on brown adipogenesis were observed in primary BAT and iWAT ex vivo, which were blocked by compound C, a selective AMPK inhibitor. These findings suggest that luteolin activates the AMPK-PPARGC1A pathway, promoting browning and thermogenesis in adipocytes to ameliorate diet-induced obesity and insulin resistance [106].

In summary, luteolin has demonstrated its potential to regulate lipid metabolism and induce browning in adipocytes, offering promising implications for addressing obesity and related metabolic disorders. In 3T3-L1 adipocytes, luteolin increased the expression of genes associated with lipid metabolism and browning, while reducing intracellular lipid accumulation. These effects were mediated through the phosphorylation of AMPK and ACC, resulting in a decrease in the expression of key adipogenic genes. Moreover, the dietary supplementation of luteolin in mice led to increased energy expenditure, elevated Ucp1 expression, and denser BAT, particularly in the mice fed with an HFD. These findings underscore luteolin’s potential to activate the AMPK-PPARGC1A pathway and promote browning and thermogenesis in adipocytes, ultimately mitigating diet-induced obesity and insulin resistance (Figure 4).

### 2.9. Menthol

Menthol, commonly used as a cooling agent in various products (e.g., foods, toothpaste, and drugs), exerts its cooling effect through the transient receptor potential melastatin 8 (TRPM8) ion channel, which senses cold stimuli. Mice lacking TRPM8 exhibited reduced sensitivity to cold temperatures [140,141].

The TRPM8 channel is involved in cold sensation and is functionally expressed in BAT. Ma et al. showed that menthol up-regulated the thermogenic capacity of adipocytes with the activation of PKA. The in vivo results also revealed that C57BL/6 mice with menthol supplementation (0.5%) under an HFD for 28 weeks showed increased body temperature upon cold exposure, whilst these effects were absent in both TRPM8 and UCP1 knockout mouse. These effects are not confined to the enhancement of thermogenic capabilities; HFD-induced obesity was prevented with the oral administration of menthol. These data showed that the activation of the TRPM8 channel through menthol supplementation increased the expression of UCP1 in brown adipose tissue, leading to an increase in thermogenesis and energy expenditure [107].

Rossato et al. investigated the effect of menthol on energy metabolism through the activation of the cold-sensing receptor TRPM8. TRPM8 was found to be expressed in cultured human WAT, and the stimulation of human white adipocytes with menthol promoted Ca^2+^ influx, resulting in the increased expression of Ucp1, mitochondrial activity, and heat generation. Additionally, menthol treatment induced morphological changes, such as browning, in the mitochondria. These findings suggested that menthol, acting through TRPM8, may serve as a browning agent, potentially modulating adipose tissue and energy metabolism in humans [48]. However, further research is needed to elucidate the specific mechanisms of menthol action in humans.

Additionally, Jiang et al. provided evidence that menthol may have beneficial effects on obesity via the promotion of browning in WAT, while simultaneously improving insulin sensitivity. In differentiated white adipocytes, menthol (500 μM) led to the activation of the TRPM8 receptor, known for its role in regulating fat metabolism. As levels of TRPM8 increased during differentiation, there was a concurrent elevation in the expression of the thermogenic genes (*Ucp1* and *Ppargc1a*) associated with browning in WAT. Furthermore, dietary menthol exhibited enhanced thermogenic capacity in vivo in DIO C57BL/6 mice. These mice were provided with an HFD with 1% menthol supplementation for 12 weeks. The menthol supplementation not only reduced body-weight gain but also improved glucose tolerance and increased the expression of thermogenic genes (Ucp1, Ppargc1a, and Prdm16) [108].

In additional in vivo experiments, the C57BL/6 mice supplemented with 0.5% menthol via an HFD for 28 weeks exhibited a 1 °C increase in body temperature when exposed to cold compared to the control group. Furthermore, the beneficial effects of menthol extended to positive impacts on insulin sensitivity and obesity improvement, as clearly demonstrated in both in vitro and in vivo studies. The oral administration of 0.5% menthol not only enhanced the mice’s thermogenic capacity but also prevented obesity induced by HFDs. These data suggest that the activation of the TRPM8 channel through menthol intake increases Ucp1 expression in BAT, leading to increased thermogenesis and energy expenditure [107].

Other studies have explored the impact of menthol supplementation on glucagon, a hormone involved in glucose metabolism, as a preventative measure against obesity induced by an HFD. The oral administration of menthol (50 and 100 mg/kg/day, 12 weeks) prevented weight gain and related diseases in HFD-fed mice. Alongside the decreased weight gain, insulin resistance, liver triglyceride levels, and adipose tissue hypertrophy were also prevented. Moreover, plasma glucagon levels were significantly increased with menthol supplementation, which is associated with increased energy expenditure and reduction in fat accumulation. In a manner consistent with the in vivo results, treating 3T3-L1 adipocytes with serum from menthol-treated animals resulted in an increased energy expenditure and expression of BAT-related genes [109]. However, blocking the glucagon receptor with a glucagon antagonist abolished the beneficial effects of menthol on body weight and glucose tolerance. Therefore, menthol may be considered to modulate glucagon levels and prevent lipid-related diseases [109].

In summary, menthol activates the TRPM8 channel and up-regulates UCP1 expression, promoting thermogenesis in both BAT and WAT in mice and inducing browning effects in WAT. Additionally, the beneficial effects of menthol extend to improving insulin sensitivity and combating obesity, as evidenced in both in vitro and in vivo studies. In animal models, menthol supplementation has been shown to reduce weight gain, increase the expression of thermogenic and energy-expenditure genes, and improve insulin sensitivity. These effects are attributed, in part, to the elevation of glucagon levels, a hormone involved in glucose metabolism. However, the intricate relationship between TRPM8 activation and elevation in calcium and PKA levels is still not clear, and further study is warranted (Figure 4). Furthermore, the contribution of other tissues in the effect of menthol on adipose browning cannot be excluded. Further studies, such as clinical trials, are needed to fully understand its effects in humans, the effective dosage, and its long-term effects. These findings underscore the multifaceted role of menthol, involving TRPM8 activation and the modulation of glucagon levels, emphasizing its therapeutic potential in addressing adipose tissue morphology, improving lipid metabolism, and addressing obesity-related issues.

### 2.10. Nobiletin

Nobiletin, a naturally occurring flavone in several citrus fruits such as *Citrus sinesis* (oranges) and *Citrus depressa*, has gained significant attention for its therapeutic potential due to its low toxicity [142]. It possesses various therapeutic qualities and biological impacts, including anti-tumor and anti-inflammatory effects and neuroprotective capabilities [143,144,145].

Lone et al. investigated the effects of nobiletin on the phenotype and browning of white adipocytes. The treatment of 3T3-L1 adipocytes with nobiletin induced enhanced expression levels of beige-specific genes, such as *Ucp1* and *Ppargc1a*. This change was accompanied with increased mitochondrial biogenesis, indicating improved energy metabolism. Moreover, the mRNA levels of *Tnfrsf9*, *Cidea*, and *TBx1*, along with increased protein levels of key transcription factors such as protein kinase A (PKA) and phosphorylated AMPK protein levels, responsible for remodeling white adipocytes, were observed with nobiletin supplementation. Furthermore, nobiletin was found to ameliorate cellular stress, as evidenced by decreased levels of reactive oxygen species and cell apoptosis. These findings suggest that nobiletin has potential as a therapeutic intervention for obesity and related diseases via improving energy metabolism and reducing cellular stress in adipocytes [110].

Kou et al. revealed nobiletin’s capacity to activate thermogenesis in mice under an HFD by shaping the gut microbiota. In C57BL/6J mice, nobiletin treatment decreased body weight compared to the control. Furthermore, along with the increased expression of thermogenic-related genes (*Ucp1*, *Prdm16*, *Cidea*), an improvement in energy expenditure was found in the nobiletin group. In addition, according to gut-microbiota analysis and metabolic profiling, mice with nobiletin consumption showed changes in gut microbiota with a reduction in Bacteroidetes’ richness and the ratio of Bacteroridetes/Firmicutes. These findings demonstrate that nobiletin may have a substantial effect in modulating gut microbiota in response to HFDs. The transplantation of the microbiota from nobiletin-fed mice to microbiota-deleted mice activated BAT activity as well as preventing HFD-induced obesity [111].

The collective findings propose that nobiletin may offer a multifaceted approach to combat obesity and related health issues. Through the promotion of the browning of adipocytes and shaping the gut microbiota to enhance thermogenesis and reduce body weight, nobiletin shows promise as a therapeutic agent for obesity management (Figure 4). However, further research is needed to elucidate the underlying mechanisms and to explore the clinical potential of nobiletin.

### 2.11. Rutin

Rutin is a natural compound found in mulberries, often utilized as a drug for capillary stabilization. It is well-known for its non-toxic nature and is clinically employed for its various pharmacological benefits, including its anti-diabetic, antioxidant, and anti-inflammatory properties [146,147,148].

In C3H10T1/2 cells, rutin treatment was found to enhance the expression of thermogenic genes (*Ucp1*, *Ppargc1a*, and *Prdm16*), as well as the processes of mitochondrial biogenesis and oxygen consumption. Similarly, supplementation with rutin (1 µg/mL) in db/db and DIO C57BL/6J male mice induced brown adipocyte activity, evidenced by increased expression levels of thermogenic genes and mitochondrial oxidative-phosphorylation (OXPHOS) protein levels. Moreover, rutin supplementation enhanced whole-body energy metabolism and maintained glucose homeostasis, while reducing adiposity in HFD conditions. Notably, rutin not only promoted BAT activity but also induced beige adipocyte formation in sWAT. In Yuan’s study, it was further demonstrated that rutin binds to Sirt1, resulting in the increased transactivation of Ppargc1a and Tfam transactivation, thereby augmenting UCP1 expression [112].

In vivo experiments conducted with male C57BL/6 mice subjected to HFD and supplemented with 0.1% rutin exhibited significantly improved blood embolic profiles and inhibited body-weight gain compared to the vehicle group under HFD. Moreover, rutin supplementation was associated with morphological changes in iWAT, resembling brown-like adipocytes. Concurrently, the thermogenic markers (*Ucp1*, *Ppargc1a*, *Ap2*, *Cebpa*, and *Pparg*) were altered, along with changes in the protein expression, indicative of the browning of white adipocytes. Consistent with the in vivo findings, in vitro experiments using 3T3-L1 cells demonstrated an up-regulation in the expression of thermogenic genes, accompanied with an increased activity of the calcium/calmodulin-dependent protein kinase 2 (Camkk2)-AMPK pathway [113].

Sheng et al. investigated the effects of mulberry leaves on obesity and revealed their potential to reduce obesity. In their study, DIO mice were supplemented with mulberry leaves, which contain significant amounts of rutin, for a period of 13 weeks. The results demonstrated that mulberry leaves effectively alleviated adiposity in DIO mice via reductions in fat accumulation, particularly in the abdominal area. Additionally, mice supplemented with mulberry leaves exhibited reduced levels of fasting blood glucose along with improved insulin sensitivity. Importantly, elevated levels of thermogenicity and thermogenic genes, including *Ucp1*, were observed in the BAT of mice treated with mulberry leaves. Moreover, the study revealed that the consumption of mulberry leaves altered the composition of the gut microbiome, leading to an increase in beneficial gut bacteria, such as Firmicutes and Proteobacteria. These findings collectively suggest that mulberry leaves have the potential to effectively reduce obesity by enhancing BAT activity through rutin supplementation [114].

The studies of Yuan, Ma, and Sheng have collectively underscored the potential of rutin, the flavonoid found in various natural sources, as a promising therapeutic agent for combating obesity and its associated metabolic disturbances. Across various experimental settings, rutin has demonstrated the ability to induce the browning of white adipocytes and activate BAT. Treatment with rutin has been shown to up-regulate thermogenic genes, enhance mitochondrial biogenesis and oxygen consumption, and promote the expression of key proteins associated with thermogenesis, in both cell cultures and animal models. These effects have been accompanied by improvements in body-weight management, glucose homeostasis, and reduced adiposity under HFD conditions. Furthermore, rutin’s impact extends to the formation of beige cells in sWAT. Mechanistically, rutin has been found to interact with Sirt1, leading to the increased transactivation of Ppargc1a and Tfam, and thereby augmenting Ucp1 expression. Additionally, rutin supplementation has been shown to induce changes in gut-microbiome composition, promoting the growth of beneficial bacteria.

In conclusion, rutin has shown promising effects in enhancing thermogenic gene expression and mitochondrial activity in cells, inducing brown adipocyte activity in obese mice, and promoting whole-body energy metabolism while reducing adiposity under HFD conditions. Moreover, the collective findings have underscored rutin’s potential as a therapeutic agent against obesity. Rutin demonstrates the ability to induce the browning of white adipocytes, activate BAT, and modulate the gut microbiome (Figure 4). However, the exact molecular mechanism underlying rutin awaits further investigation, and additional research is required to fully elucidate rutin’s clinical potential in anti-obesity interventions and to explore its mechanisms of action in greater detail.

## 3. Conclusions

In conclusion, this review summarizes the relationship between natural compounds and adipose thermogenesis, offering a comprehensive overview of the potential mechanisms involved. The evidence presented suggests that certain bioactive compounds can induce adipose browning through various signaling pathways, including the activation of AMPK/SIRT1/PPARGC1A, the stimulation of TRPM8, the β3-AR receptor, or TGR5, and the interaction with key regulators like PRDM16, PPARGC1A, and PPARA/G. These findings underscore the therapeutic potential of natural compounds in combating obesity and related metabolic disorders via the promotion of thermogenesis in adipose tissues.

It is important to recognize the limitations inherent in the current body of research on natural compounds and their effects on adipose thermogenesis. While studies involving berberine and menthol have shown promising results in both in vitro and in vivo settings with relevance to humans (Table 1), the scope of research on these compounds remains limited. For example, while berberine supplementation has demonstrated effects on BAT volume and activity in NAFLD patients, the optimal dosage and mechanism of action in humans are not fully understood. Similarly, although menthol has shown beneficial effects on thermogenic gene expression and mitochondrial function in human adipocytes, further research is needed to elucidate their therapeutic potential and establish optimal dosing regimens.

Furthermore, numerous studies depend on in vitro (Table 2) and animal models (Table 3) to investigate the impacts of natural compounds, necessitating careful consideration when extrapolating these findings to human physiology. The specific dosage, duration, and potential interactions of natural compounds are not adequately understood, posing challenges in establishing standardized recommendations for therapeutic applications for humans. Additionally, the individual responses to these compounds may vary, highlighting the importance of personalized approaches in harnessing their potential benefits. Overall, while the current research provides valuable insights into the potential therapeutic effects of natural compounds on adipose thermogenesis, further studies are needed to address these limitations and fully elucidate their clinical relevance and applicability.

In conclusion, while the current body of evidence underscores the exciting possibilities of natural compounds, caution is advised in extrapolating any findings to clinical applications. Future research endeavors should focus on overcoming these limitations to provide a more nuanced understanding of the interplay between natural compounds and adipose thermogenesis, paving the way for safer and more effective therapeutic interventions in the realm of metabolic health.

This comprehensive review provides insights into the current state of knowledge regarding the interplay between thermogenesis and specific natural compounds (Figure 5). While the evidence supporting the thermogenic effects of these compounds is promising, further research is needed to elucidate the underlying mechanisms and optimize their potential as therapeutic interventions for metabolic disorders. Understanding the intricate relationship between natural compounds and thermogenesis holds promise for developing innovative strategies to combat obesity and related metabolic diseases.

## Figures and Tables

**Figure 1 ijms-25-04915-f001:**
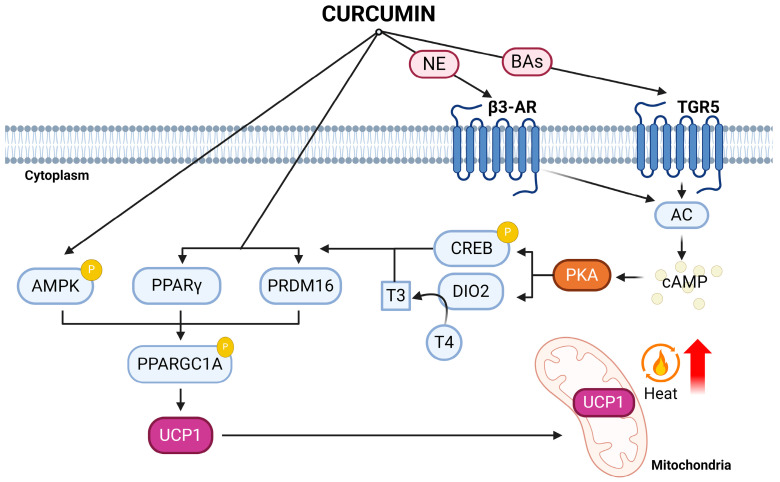
The mechanism of action of curcumin on adipose browning. Curcumin induces adipose browning through activation of UCP1 via β3-AR, TGR5, and AMPK phosphorylation. Curcumin up-regulated PPARγ, PRDM16, and phosphorylated AMPK, which in turn activated Ppargc1a and UCP1 [35,37,39]. Additionally, curcumin exerted beneficial effects by altering gut-microbiome symbiosis and bile acids. BAs are known to activate TGR5, which triggers the production of cAMP. Activation of TGR5 through curcumin supplementation induced the phosphorylation of CREB, up-regulating UCP1-dependent thermogenesis [49]. Curcumin was also shown to increase the level of norepinephrine in serum upon supplementation [40,42]. Norepinepherine, up-regulated by curcumin, activates β3-AR, leading to the induction of DIO2 expression. Increased DIO2 enhances the conversion of T4 to T3 within adipocytes. Subsequently, T3 activates the expression of genes involved in thermogenesis, PRDM16, PPARγ, PPARGC1A, and UCP1 [40,42]. Overall, curcumin’s mechanism of action involves the activation of key regulators such as UCP1 through β3-AR, TGR5, and AMPK phosphorylation, alongside modulation of PPARγ, PRDM16, and AMPK phosphorylation. Ppargc1a activates mitochondrial biogenesis while UCP1 increases thermogenesis. The diagram includes key components such as norepinephrine (NE), bile acids (BAs), adenylyl cyclase (AC), 5′ AMP-activated protein kinase (AMPK), beta-3 adrenergic receptor (β3-AR), cAMP-response element-binding protein (CREB), iodothyronine deiodinase 2 (Dio2), phosphorylation (P), peroxisome proliferator-activated receptor γ coactivator-1α (PPARGC1A), protein kinase A (PKA), PR domain zinc finger protein 16 (PRDM16), peroxisome proliferator-activated receptor gamma (PPARG), thyroxine (T4), triiodothyronine (T3), Takeda G protein-coupled receptor 5 (TGR5), and uncoupling protein-1 (UCP1).

**Figure 2 ijms-25-04915-f002:**
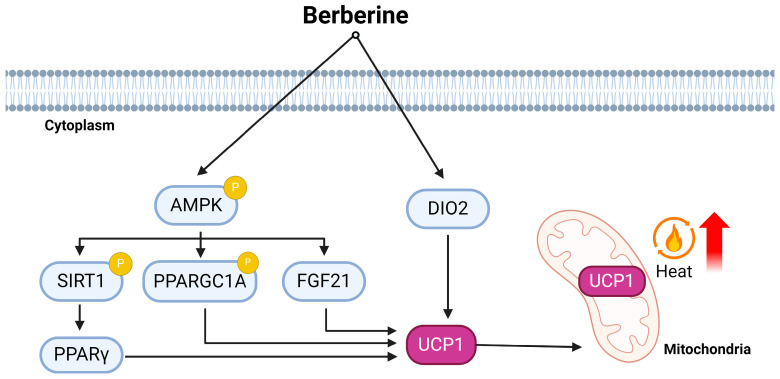
The mechanism of action of berberine on adipose browning. Berberine induces thermogenesis and adipocyte browning via stimuli of FGF21 and DIO and activating the AMPK/SIRT1/PPARGC1A pathway. Berberine phosphorylated AMPK, which in turn activated SIRT1. Phosphorylated SIRT1 promotes the deacetylation and activation of PPARGC1A and the induction of PPARγ, leading to the induction of UCP1 expression and the subsequent enhancement of thermogenesis in brown adipose tissue [54,55]. In addition, berberine was also shown to regulate UCP1 expression by augmenting FGF21 levels through AMPK phosphorylation [56]. On the other hand, berberine supplementation in DIO mice was shown to increase UCP1 alongside DIO2 expression levels; however, further study is needed to elucidate the exact mechanism of action [62]. This signaling cascade resulted in an increase in thermogenic markers, along with an increase in UCP1. The diagram includes key components such as 5′ AMP-activated protein kinase (AMPK), peroxisome proliferator-activated receptor gamma (PPARG), iodothyronine deiodinase 2 (Dio2), phosphorylation (P), fibroblast growth factor 21 (FGF21), iodothyronine deiodinase 2 (DIO2), peroxisome proliferator-activated receptor γ coactivator-1α (PPARGC1A), sirtuin-1 (SIRT1), and uncoupling protein-1 (UCP1).

**Figure 3 ijms-25-04915-f003:**
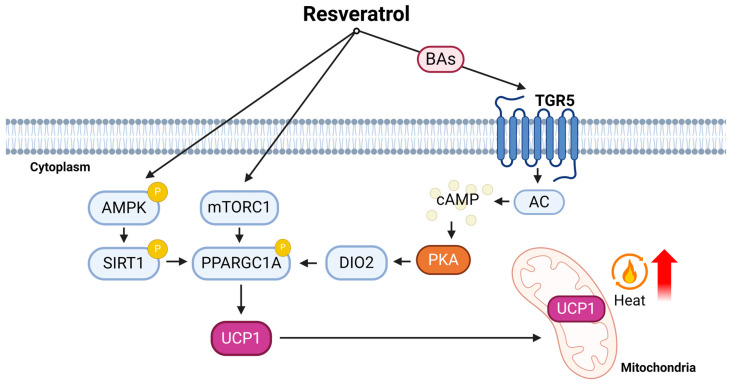
The mechanism of action of resveratrol on adipose browning. Resveratrol induces adipose browning via the activation of the mTORC1, AMPK-SIRT1, and TGR5—PPARGC1A pathways. Upon the activation of resveratrol, AMPK phosphorylates and activates SIRT1, a nicotinamide adenine dinucleotide (NAD+)-dependent deacetylase. Phosphorylated SIRT1, in turn, exerts its enzymatic activity to deacetylate and activate PPARGC1A, a master regulator of mitochondrial biogenesis and UCP1 expression. Subsequently, activated PPARGC1A promotes the transcription of UCP1, leading to the induction of thermogenesis in adipose tissue [74,77]. Additionally, resveratrol increases the concentration of bile acid and lithocholic acid, which in turn activates TGR5 receptor. The activation of the TGR5 receptor stimulates AC, resulting in elevated levels of cAMP. It then activates PKA, which phosphorylates and activates DIO2. Activated DIO2 enhances the expression of PPARGC1A, promoting the transcription of UCP1, facilitating thermogenesis and energy dissipation in brown adipose tissue [85]. The diagram includes key components such as bile acids (BAs), 5′ AMP-activated protein kinase (AMPK), cyclic adenosine monophosphate (cAMP), type 2 iodothyronine deiodinase (DIO2), adenyl cyclase (AC), the cAMP-response element-binding protein (CREB), phosphorylation (P), peroxisome proliferator-activated receptor γ coactivator-1α (PPARGC1A), sirtuin-1 (SIRT1), Takeda G protein-coupled receptor 5 (TGR5), and uncoupling protein-1 (UCP1).

**Figure 4 ijms-25-04915-f004:**
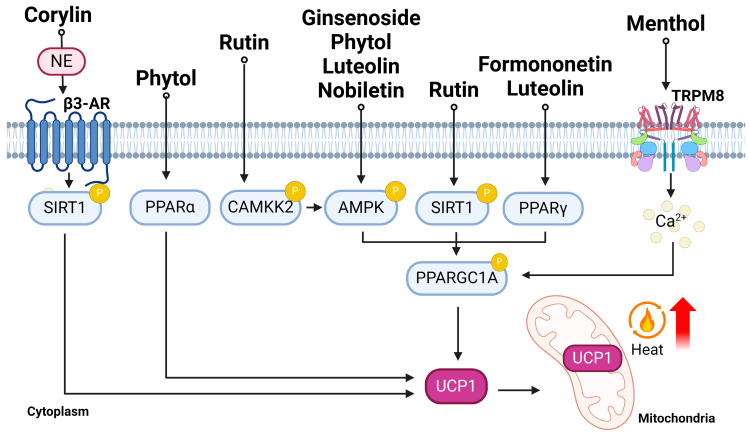
A diagram illustrating the potential mechanisms of corylin, formononetin, ginsenoside, phytol, luteolin, menthol, nobiletin, and rutin-induced adipose browning. The signaling pathways responsible for the up-regulation of UCP1 and its transcriptional regulators are depicted. Corylin induces adipose browning by activating SIRT1 and β3-AR; however, the underlying mechanisms are unknown [101]. Formononetin [102], ginsenoside [95,96,99,100], phytol [103,104], and luteolin [105,106] induce adipose browning by activating the AMPK-PPARGC1A pathway. Menthol stimulates adipose browning through the AMPK-PPARGC1A pathway, as well as by stimulating TRPM8 [48,107,108,109]. Nobiletin [110,111] and rutin [112,113,114] stimulate adipose browning by up-regulating PPARGC1a and UCP1; however, the signaling components are yet to be elucidated. The diagram includes key components such as adenylyl cyclase (AC), 5′ AMP-activated protein kinase (AMPK), activating transcription factor 2 (ATF2), beta-3 adrenergic receptor (β3-AR), cAMP-response element-binding protein (CREB), iodothyronine deiodinase 2 (Dio2), phosphorylation (P), peroxisome proliferator-activated receptor γ coactivator-1α (PPARGC1A), peroxisome proliferator-activated receptor α (PPARα), calcium/calmodulin-dependent protein kinase 2 (CAMKK2), sirtuin-1 (SIRT1), transient receptor potential cation channel subfamily M member 8 (TRPM8), and uncoupling protein-1 (UCP1).

**Figure 5 ijms-25-04915-f005:**
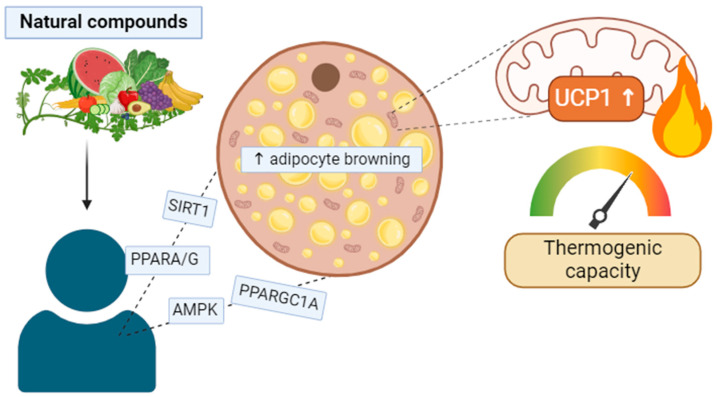
A schematic diagram illustrating the browning effects of natural compounds on adipose thermogenesis regulation. The consumption of natural compounds improves and enhances adipose browning through signaling pathways such as SIRT1, PPARA/G, AMPK, and PPARGC1A. The up-regulation of these markers results in increased levels of UCP1, in other words, thermogenic capacity, which could be an effective strategy in regulating metabolic health.

**Table 1 ijms-25-04915-t001:** Bioactive compounds’ effects in humans or other organisms.

Bioactive Compounds	Model (Treatment Dosage)	Biomarkers	Ref.
Berberine	10 NAFLD patients	Increased volume and activity of BAT. Increased basal oral temperature.	[62]
Resveratrol	C. elegans	Extended lifespan.	[149]
Menthol	Human adipocytes	Simulated Ca^2+^ influx to elevate UCP1 expression and mitochondrial activity. Brown-like morphological changes in mitochondria.	[48]

**Table 2 ijms-25-04915-t002:** Bioactive compounds effect in vitro.

Bioactive Compounds	Model (Treatment Dosage)	Biomarkers	Ref.
Curcumin	3T3-L1 and iWAT primary adipocytes from rat (20 µM)	Ucp1, Ppargc1a, Prdm16, C/EBPb, Cidea, Fgf21	[35]
	3T3-L1 adipocytes (10, 20, 35 μM curcumin)	Ucp1, Prdm16, Pparg, Ppargc1a	[37]
Berberine	C3H10T1/2 mesenchymal stem cells	Ucp1 AMPK-Ppargc1aa pathway	[55]
	3T3-L1 adipocytes HIB1b cell	Ucp1, Sirt1, PPARA AMPK-SIRT1 pathway	[54]
	C3H10T1/2 mesenchymal stem cells	Fgf21 level AMPK signaling pathway	[56]
Corylin	3T3-L1 adipocytes	Sirt1 ß3-AR-dependent pathways	[101]
Resveratrol	Primary stromal vascular cells	Ucp1, Prdm16, Ppargc1a, PDH, CytoC AMPK1-dependent manner	[74]
Formononetin	Primary iWAT cells (1 μM)	Ucp1, Prdm16, Ppargc1a, Cidea, Dio2 PPARγ-UCP1 pathway	[102]
Ginsenoside	3T3L1 (10 μM Ginsenoside Rb1)	Ucp1, Ppargc1a, Prdm16 PPARγ dependent pathway	[95]
	3T3-L1 (20 μM Ginsenoside Rb1)	Ucp1, Ppargc1a, Pppara Increase in the level of sirtuins (SIRT1 and SIRT3)	[96]
	Differentiated C3H10T1/2, primary iWAT, 3T3-L1 (10 μM Ginsenoside Rb2)	Ucp1, Prdm16, Ppargc1a, Cidea, Dio2, Ucp1, Fgf21, ATP syn, Cox8β AMPK-Ppargc1a-Ucp1 pathway	[99]
	3T3-L1 adipocytes (20, 40 μM Ginsenoside Rb3)	Ucp1, Prdm16, Ppargc1a, Cidea, Dio2 AMPK-PPARGC1A-UCP1 pathway	[100]
Phytol	3T3-L1 adipocytes (100 μM)	Ucp1, Prdm16, Ppargc1a, Cidea, Elovl3, AMPKa pathway	[103]
	3T3-L1 adipocytes (50 μM)	Ucp1, Prdm16, Ppargc1a, Cidea	[104]
Luteolin	3T3-L1 adipocytes (10, 20, 40 μM)	Ucp1, Ppargc1a Increased phosphorylation of AMPK and ACC	[105]
Menthol	Differentiated mouse sWAT adipocytes (500 μM)	Activation of TRPM8 receptor Ucp1, Ppargc1a	[108]
Nobiletin	3T3-L1 adipocytes	Ucp1, Ppargc1a Increased PKA and AMPK protein level	[110]
Rutin	C3H10T1/2 cells	Ucp1, Ppargc1a, Prdm16 SIRT1-Ppargc1a-UCP1 pathway	[112]
	3T3-L1 adipocytes	Ucp1, Ppargc1a, Ap2, Pparg CaMKKβ-AMPK pathway	[113]

**Table 3 ijms-25-04915-t003:** Bioactive compounds’ effects in vivo.

Bioactive Compounds	Model (Dosage, Duration)	Molecular Target	Metabolic Actions	Ref.
Curcumin	HFD-fed male C57BL/6 (50, 100 mg/kg/day by gavage, 50 days)	Ucp1, Ppargc1a, Prdm16, Dio2, PPARA, Cidea, Elovl3, Nrf1, mtTfa, ATPsyn in iWAT	Decreased body and fat mass. Improved cold tolerance. Increased plasma norepinephrine level.	[40]
	HFD-fed male C57BL/6J mice (1% curcumin in chow, 18 weeks)	Ucp1, PPARA, Pparg, Ppargc1a	Increased CO_2_ production, O_2_ consumption, and energy expenditure. Increased thermogenic capacity.	[39]
	HFD-fed male C57BL/6J mice, UCP1^−/−^ and TGR mice (100 mg/kg body weight curcumin)	Ucp1 cAMP-PKA-CREB-UCP1 pathway	Decreased body weight. Increased thermogenic capacity. Changes in gut microbiota.	[49]
	3-week-old rat litter (2% curcumin supplementation)	Ucp1 β3 adrenergic receptor mRNA expression	Decreased bodyweight. High lean mass. Increase in plasma norepinephrine level.	[42]
	Male C57BL/6 (0.2% curcumin supplementation)	Ucp1, Ppargc1a	Improved insulin sensitivity. Decreased fat mass. Decreased gut microbiota.	[150]
Berberine	Six-week-old male C57BL/6J mice (1.5 mg/kg/day BBR, i.p. injection, 6 weeks)	Ucp1, Ppargc1a, Nrf1	Increased energy expenditure, thermogenesis, and BAT activity.	[62]
	Obese C57BLKS/J-Leprdb/Leprdb (db/db) male mice (5 mg/kg per day, i.p. injection, 4 weeks)	Ucp1, Cidea, Cox8b, Isdp5	Increased whole-body energy expenditure. Higher oxygen consumption. Higher CO_2_-production rate. Limits weight gain.	[55]
	C57BL/6 mouse and SIRT^−/−^ transgenic mouse (25 mg/kg, 100 mg/kg BBR supplementation)	Ucp1, Sirt1, Pparg, Ppargc1a, Dio2	Decrease in body weight. Increased energy expenditure.	[54]
	ddY mice (5 mg/kg/day BBR i.p. injection)	Fgf21, AMPK	Increase in core-clock component brain and muscle Arnt-like 1 (Bmal1). Knockdown of Bmal1 gene left FGF21 unaltered.	[56]
Corylin	HFD DIO C57BL/6 (9 weeks)	SIRT1-β3-AR dependent pathway	Decrease in visceral fat mass. Decrease in the size of WAT adipocytes. Increase in energy expenditure.	[101]
Resveratrol	Female CD1 mice (0.1% resveratrol)	Ucp1, Prdm16, Ppargc1a, Pdh, Cyto C	Increase in oxygen-consumption rate. Phosphorylation of AMPK1.	[74]
	DIO C57BL/6J offspring	Ucp1, Prdm16, Ppargc1a	Increase in energy expenditure. Increase in insulin sensitivity. Enhanced browning of adipocytes in the offspring.	[80]
	Db/db mice	Ucp1, Cidea, Prdm16, Ppargc1a	Increase in BAT and decrease in size and mass of WAT. Reorganization of the gut microbiota.	[85]
Formononetin	DIO mice (50 mg/kg/day)	Ucp1, Cidea, Ppargc1a, Prdm16, Dio2	Increased multilocular structure of adipocytes. Decrease in fat mass.	[102]
Ginsenoside	Dio C57BL/6 mice (40 mg/kg/day Ginsenoside Rb2)	Ucp1 AMPK phosphorylation	Decrease in body weight. Enhanced insulin sensitivity. Increase in energy expenditure.	[99]
Phytol	C57BL/6J mice (500 mg/kg/daily, via oral gavage, 7 weeks)	Ucp1, Ppargc1a, Prdm16, Cyto C AMPKa signaling pathway	Decrease in body-weight gain. Multilocular droplets in iWAT.	[103]
Luteolin	C57BL/6 mice (0.01% luteolin supplement, HFD)	Ppargc1a, PPARA, Cidea and Sirt1 AMPK-Ppargc1a signaling pathway	Increased energy expenditure, oxygen-consumption rate. Denser BAT.	[106]
Menthol	DIO C57BL/6 mouse (1% menthol supplementation, 12 weeks)	Ucp1, Ppargc1a, Prdm16	Reduced body-weight gain. Improved glucose tolerance.	[108]
	C57BL/6 mice (0.5% menthol supplementation, 28 weeks)	Ucp1	Increased body temperature by 1 °C upon cold exposure. Decrease in body weight.	[107]
	C57BL/6 mice (50 and 100 mg/kg/day, 12 weeks)	Ucp1, Ppargc1a, Prdm16	Decrease in weight. Prevention of adipose tissue hypertrophy. Increase in plasma glucagon level.	[109]
Nobiletin	C57BL/6 (50 mg/kg body weight, 12 weeks)	Ucp1, Ppargc1a, Prdm16, Cidea	Changed gut microbiota with reduction in Bacteroidetes richness.	[110]
Rutin	C57BL/6J (1 ug/mL rutin supplementation)	Ucp1, Ppargc1a, Prdm16	Increased whole-body energy metabolism. Maintained glucose homeostasis.	[112]
	C57BL/6 mice (0.1% rutin supplementation	Ucp1, Ppargc1a, Ap2, Pparg	Multi-locular droplets in iWAT.	[113]
	C57BL/6 (mulberry leaf supplementation)	Ucp1, Ppargc1a	Increase in beneficial gut bacteria. Reduced level of blood glucose and improved insulin sensitivity.	[114]

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
