# Peer review of "Unveiling the Potential of Natural Compounds: A Comprehensive Review on Adipose Thermogenesis Modulation"

_ijms, 2024, doi:10.3390/ijms25094915_

Round 1
Reviewer 1 Report (Previous Reviewer 1)
Comments and Suggestions for Authors
This review has been greatly improved since the first version. My only comment would be that the word "determination" throughout the introduction be switched to the word "commitment". For example: The authors wrote" The transcriptional regulation of adipogenesis involves two main phases: determination and terminal differentiation." Please use the word commitment.
Comments on the Quality of English Language
The English has been greatly improved, however, there are still too many grammatical and sentence structure errors throughout the manuscript. Too many to list.
Author Response
Please see the attachment.

Reviewer 2 Report (Previous Reviewer 2)
Comments and Suggestions for Authors
The review „Unveiling the Potential of Natural Compounds: A Comprehensive Review on Adipose Thermogenesis Modulation” before publication need to be revised, here are main issue:
-lines 130 – 136 no citation, here should be [38] citation , before [37], now is confusing, suggesting all the tekst belongs to publication 37,
-after lines 169 – 179 should be citation [40], now suggest that [36] is the author of statement, it is confusing,
- line 278 – 279 – no citation
- line 281-282 – no citation
- line 290 - incomprehensible sentence
- line 319 – 324 – lack of citation, suggesting that [63] is the author of the statement, confusing
- line 413, 414, 415 correct pre-vent, de-creasing ect.
- standardize gene names, sometimes it is in italics, sometimes not,
- line 487 - lack of citation
- line 492 - wrong citation [90], the statement from line 491 and citation 89 and 90 should be removed
- line 632 – wrong citation 119, there is no mention about phytol
- line 717-727- lack of citation
- line 767 – lack of citation
Author Response
Please see the attachment.

Reviewer 3 Report (Previous Reviewer 3)
Comments and Suggestions for Authors
no further comments
Comments on the Quality of English Languagemoderst corrections needed
Author Response
Please see the attachment.

Reviewer 4 Report (Previous Reviewer 4)
Comments and Suggestions for Authors
The manuscript has been improved, however I do not feel fully satisfied. Still no inofrmation is provided on the search methodology, and also the limitations of the studies need to be extended.
Round 2
Reviewer 2 Report (Previous Reviewer 2)
Comments and Suggestions for Authors
No comments
This manuscript is a resubmission of an earlier submission. The following is a list of the peer review reports and author responses from that submission.
Round 1
Reviewer 1 Report
Comments and Suggestions for Authors
This review offers an overview of natural food compounds activating brown adipocytes and explores their potential therapeutic mechanisms against obesity. However, there are issues with accurate and consistent in-text citations and references, with references not in correct order and some not matching the corresponding articles. Throughout the article, there is text with missing references or no references at all. Sometimes the references are in brackets [], and in other places author names are used (Han et al., 2021). Large sections of the manuscript are written from just a single reference. The impact of the article is diminished due to poor referencing and poor writing quality, making it challenging to assess the manuscript's accuracy. The article needs to be more concise. Additionally, unclear sentences and inadequate explanation of adipocyte concepts further detract from the quality of the manuscript. Here are just a few examples from the first 2 pages of the article:
1) WAT is known to contain beige adipocytes on occasion, which 48 are frequently descended from white adipocytes [3]. It is much more complex than this. They can be induced from both progenitor cells and from mature whitened beige adipocytes.
2) Sentence 42: by cold or during activity. What activity?
3) Sentence 40: is essential to store more energy than triglycerides. This sentence doesn’t make sense. Store more energy in the form of triglycerides?
4) Sentence 44 needs a reference.
5) Due to their resemblance in shape and function to brown adipocytes, 45 beige adipocytes have become known. However, as their growth was more closely related 46 to white adipose tissue, they were given the name "beige adipocyte," referring to the color 47 between white and brown. WAT is known to contain beige adipocytes on occasion, which 48 are frequently descended from white adipocytes [3]. This is not clear or factual. Needs a better description for what beige adipocytes are and the browning process.
6) Statements such as the following are not factual. “curcumin can inhibit the hypertrophy of white fat cells and promote the 89 transition into BAT.” White fat cells turn into beige fat cells, not into BAT (brown adipose tissue).
7) Furthermore, compared to control group, curcumin fed DIO group 90 showed increased WAT. This doesn’t make sense.
8) expression of M1-like pro-inflammatory cytokines and gene markers (Ucp1, Ppargc1a) in 98 both adipocytes and macrophages. Ucp1 isn’t expressed on macrophages.
9) Curcumin also increased the level of norepi- 112 nephrine, leading to the conclusion that it upregulates the expression of ß3-AR, which is 113 responsible for over [2]. Sentence doesn’t make any sense.
Comments on the Quality of English Language
Not acceptable, needs major work.
Reviewer 2 Report
Comments and Suggestions for Authors
The manuscript before submission should be carefully checked out to see if references are cited accordingly, and there are many wrong citations and two different citation styles. Wrong citation or missing citations: Introduction- 3 and 4 should be cited inversely; lines 84, 94, 102, 114, 122,126, 134,138, 15, 158, 18, 191, 199, 204, 205, 210and 213(lack), 219, 224, 294, 297(lack), 309, 319 -wrong citation; I did not check further because manuscript should be first improved then submitted
Reviewer 3 Report
Comments and Suggestions for Authors
In general, this is a well written manuscript, which may be improved by helping the reader by correcting the following points:
1. All abbreviations should be defined at first time use and then consistently applied. This applies also to gene/protein names.
2. A reference to the two figures and three tables is largely missing. Moreover, the Figures and tables should be presented far earlier in the manuscript.
3. The described data are largely derived from experiments with rodents. Please emphasize more clearly, which conclusions are also valid for humans.
4. Please try to summarize the overall conclusion of this review by a summarizing figure, which may act in parallel also as a graphical abstract.
Comments on the Quality of English LanguageModerate corrections needs.
Reviewer 4 Report
Comments and Suggestions for Authors
The presented manuscript focused on the review of the impact of some natural compounds on adipose thermogenesis modulation. The manuscript is well writtem and with spome nice pictures, illustrating the mechanism of the action of the described compounds. The paper in overall seems to be interesting, however I have some points to be expalined or corrected, which are as follows:
- no information is provided on the process of the selection of the compounds, included in the manuscript, nor on the data research details. The authors should justify the choice of those particular compounds, but also what databases were searched, with what keywords, what were the exclusion criteria for the articles etc.
- a paragraph with some critical comments describing the limitations of the studies cited should be welcomed - this may help the reader to evaluate the effictiveness of the compounds described